# Some of Problems of Direction Finding of Ground-Based Radars Using Monopulse Location System Installed on Unmanned Aerial Vehicle

**DOI:** 10.3390/s20185186

**Published:** 2020-09-11

**Authors:** Adam Rutkowski, Adam Kawalec

**Affiliations:** Faculty of Electronics, Military University of Technology, ul. Gen. Sylwestra Kaliskiego 2, 00-908 Warsaw, Poland; adam.rutkowski@wat.edu.pl

**Keywords:** direction finding (DF), microwave emitter (ME), angle of arrival (AOA), direction of arrival (DOA), monopulse direction finding (MDF), microwave phase discriminator (MPhD), instantaneous frequency measurement (IFM)

## Abstract

Locating active radars in real environmental conditions is a very important and complex task. The efficiency of the direction finding (DF) of ground-based radars and other microwave emitters using unmanned aerial vehicles (UAV) is dependent on the parameters of applied devices for angle location of microwave emitters, and on the construction and modes of operation of the observed transmitting antenna systems. An additional factor having the influence on DF of the radar, when are used systems installed on the UAV, is the rotation of the antenna of a radar. The accuracy of estimation of direction of any microwave transmitter is determined by the terrain properties that surround the transmitter and the objects reflecting microwave signals. The exemplary shapes of the radar antenna patterns and the associated relationships with the probability of remotely detecting the radar and determining its bearings are described. The simulated patterns of the signals received at an emitter-locating device mounted on a UAV and the expected results of a monopulse DF based on these signals are presented. The novelty of this work is the analysis of the DF efficiency of radars in conditions where intense multi-path phenomena appear, and for various amplitudes and phases of the direct signal and multi-path signals that reach the UAV when assuming that so-called simple signals and linear frequency modulation (LFM) signals are transmitted by the radar. The primary focus is on multi-path phenomenon, which can make it difficult, but not entirely impossible, to detect activity and location of radar with a low-flying small UAV and using only monopulse techniques, that is, when only a single pulse emitted by a radar must be sufficient to DF of this radar. Direction of arrival (DOA) algorithms of signals in dense signal environment were not presented in the work, but relevant suggestions were made for the design of such algorithms.

## 1. Introduction

The angular location of active radars can be determined using ground-based installations or mobile systems, or applying aerial vehicles equipped with direction finding (DF) equipment. The latter issue exploits small, unmanned aerial vehicle (UAV), and is very useful in many civil and military applications. Unmanned aerial vehicles, such as drones or gliders, can aviate both at relatively high and small altitudes with both relatively high and low velocities, while also remaining stationary in the air. Some types of UAVs can have small dimensions, so onboard antenna systems for the DF device should be adequately small. This means that the antenna system should contain a limited number of single antennas located a small distance from each other. When the DF device installed on the UAV observes a radar, it receives not only the direct signal but signals that reflect from the ground or other reflecting objects as well. Very often, signals reflected from local objects are stronger than signals reflected from the ground. This is very similar to what happens when radars observe low-altitude targets [1,2]. This effect causes the estimated bearings of the radar to be erroneous. What’s more, the rotation of the radar antenna causes the strength of the signal received by the UAV to change quickly, and even the signal may fade at times. This effect intensifies the problem of accurately locating the radar in angle space by means of DF device mounted on UAV. 

There are many various methods of direction finding and localizing of objects. One of them is described in [3,4,5], for example. This solution does not rely on microwave signals. It is a tracking system that utilizes inertial sensors working in smartphones. Such a system could be used as an on-board support for tracking the position of a UAV equipped with microwave passive direction finding systems. In [6], the interferometric network of direction of microwave source estimation is described. This network cooperates with the eight omnidirectional antennas arranged on a circle and single omnidirectional antenna placed at the center of this circle. The signal angle of arrival (AOA) estimation is based on comparing the phases of the signals from each antenna. An additional problem with direction finding is the ability to resolve the closely spaced targets. The method of resolving targets by monopulse radar using Levenberg-Marquardt is presented in [7]. This solution provides high-angle measuring accuracy for one and two targets by performing a series of complex computer calculations. The angular separation of the two closely located targets was obtained in [8], using a radar diagonal difference channel in the monopulse radar with a four-element antenna system. The solution with four independent channels used to separate the directions of two closely spaced microwave transmitters is also presented in [9,10]. Another interesting way to estimate the direction of multiple targets is presented in [11]. In this work, it is proposed to use the so-called independent component analysis (ICA) first, followed by the monopulse phase comparison technique to determine the direction of each target. The accuracy of estimating the direction of microwave sources can be improved using several DF devices working simultaneously, or by measuring the direction utilizing a single DF device successively at different points in the space surrounding the emitting source [12]. The known direction finding solutions mentioned above are very effective and accurate, due to a great computational effort, among other reasons. However, for our projects, we need solutions with simplest possible construction of microwave blocks and the lowest computational complexity. 

This study is dedicated to finding a way to reduce errors from estimating the direction of a radar despite the existing distortion features mentioned above. The solution of the problem is searched through the analysis of various distinctive features that exist when are received microwave signals in a dense signal environment, where many reflecting objects appear and/or many emitters work simultaneously. Therefore, temporary values of the resultant signals received by the DF antennas installed on UAV were investigated. Afterwards, these signals were used to estimate the angular location of the radar using monopulse methods. The results of such estimations formed the data sets that contain both correct and incorrect bearing values. The novelty and primary task of the work are identification, which portions of each plot relate to the accurate direction measurements of the radar. These parts of plots will be used to help the systems cooperating with DF device to make further decisions, and the rest of information can be neglected. 

The purpose of this paper is to show some important phenomena related to determining the direction of radars, and not to describe the algorithms that are worked on. This time, the object of interest was neither clutter nor multipath phenomena, but the consequences of their appearance, i.e., that the radar emits a single pulse, but the DF system installed on the UAV can receive several pulses or a pulse-like signal longer than that transmitted by observed radar. The 2-antenna geometry working only in elevation plane was chosen for this work, not a multi-element 2D antenna system, because the main task of the work was to clearly show important phenomena and their properties. Only after analyzing these phenomena it is possible design effective DF algorithms useful in a dense signal environment. In addition, on some small UAVs and in some applications there is no room for using the extended DF antenna array. 

## 2. Radar Antenna Radiation Pattern

The exemplary shapes for the directional pattern of a radar transmitting antenna are presented in Figure 1 and Figure 2. The main beam and side lobes [1] are visible in these patterns. The differences in power along the main beam direction and the side lobes maxima are approximately 30–40 dB [1,13]. The minimums (nulls) exist between adjacent lobes, which have transmitted power levels that are very low, though not zero in practice. 

The main beam of a transmitting antenna pattern of a radar designed for airspace observations has usually the shape of a vertical fin. This type of main beam is wide in the elevation plane (Figure 1) and relatively narrow in the azimuth plane (Figure 2). The power of the signal received by the DF device on the UAV depends on the power radiated from the observed transmitter, and is also dependent on the UAV altitude. 

For the case shown in Figure 1, UAV1 is illuminated by the main beam at a relatively large power level, which is close to maximum of the antenna pattern. However, the signal from the same radar reaching UAV2, which is traveling at the same angle (in azimuth plane) and distance but at lower altitude (elevation), has a much lower signal. This is because the signal comes from a region of the radar transmitting antenna gain curve which is placed near its first minimum. 

Some radars observe a narrow sector of space, so its main beam is fixed and directed towards a specific region of interest. If the radar observes a wide segment of space, then its radiation pattern is scanned either electronically or through mechanical rotation. Such variants of radar antenna systems are shown in Figure 2. At the moment *t*_1_ (Figure 2a), the DF device on the UAV easily detects and locates the radar, because the UAV receives the signal with a sufficiently high power level from the main beam. However, the antenna at moment *t*_2_ has shifted (Figure 2b), so the UAV is in a null (minimum), and will receive a relatively weak signal. In such conditions, radars are localized with significant errors, or may not even be detected. Naturally, in the next moment, as a consequence of antenna rotation, the UAV will be in an antenna side lobe, so a stronger signal will be detected. After a full rotation, the UAV will again be in the antenna main beam, so the DF device will receive a strong signal. This is when the angular location of radar can be the most precisely estimated.

## 3. Multipath Phenomenon

Multipath phenomena in radiolocation occur very often [2]. In this case, signals from the emitter reach the receiver through multiple pathways. The multipath intensity depends, among other things, on the radar antenna height H_R_ and on the UAV altitude H_P_. The multi-path effect is illustrated in Figure 3. 

The reflection and incidence angles of microwave signal off a surface are equal each other [2], so the propagation tracks of reflected signals can be very diverse. In Figure 3, the direct signal propagates towards point P only through the air and travels a distance R. The signals from the side lobes are initially aimed towards the ground, and are reflected and propagate in various directions depending on the incidence angle. The reflected signals that reach to the same point P are called multipath signals. The propagation routes of these signals are longer than for the direct signal. The differential time of arrival of the multipath signal depends on the distance of the reflection point from the radar. If the reflection occurs at a distance L*_Ri_* from the radar, then the multipath signal travels a path length of *L_MPi_*, as given by Equation (1).
*L_MPi_* = *R_i_*_1_ + *R_i_*_2_, (1)
where: *i*—number of reflection point (*i* = 0, 1, 2, 3).

As seen in Figure 3, the lengths *L_MPi_* are different for each path. As the times of arrival vary, their phases at point P will be adequately different. Moreover, due to the various reflective properties of the multipath reflection points, both the powers and phases of the signals received at point P can be diverse. 

In general, the so-called resultant signal *u(t)* received at a given point P of the space surrounding the radar shown in Figure 3 is the vectoral sum of the direct signal *u_D_(t)* and the multipath signals *u_M_(t)*. This sum can be expressed by Equations (2)–(4).
*u_D_(t)* = *U_D_* ⋅ sin[2 ⋅ π ⋅ *f*(*t*) + Φ*_D_*(*t*) + Φ_0*D*_], (2)
*u_M_(t)* = ∑_m = (0 ÷ *N*)_ {*U_m_* ⋅ sin[2 ⋅ π ⋅ *f*(*t* + *t_m_*) + Φ*_m_*(*t* + *t_m_*) + Φ_0*m*_]}, (3)
*u(t)* = *u_D_* (*t*) + *u_M_* (*t*), (4)
where: *U_D_*—amplitude of the direct signal, *t*—time, *f*(*t*)—temporary value of the frequency, Φ*_D_*(*t*)—temporary phase of the direct signal, Φ_0*D*_—initial phase of the direct signal, *N*—quantity of multipath signals reflected from the ground and received at point P, *m*—number of multipath signal reflected from the ground (*m* = 0, 1, 2, …, *N*), *t_m_*—temporary delay of the reflected signal *m* in relation to the direct signal, *U_m_*—amplitude of the reflected signal *m*, Φ*_m_*(*t*)—temporary phase of the reflected signal *m*, and Φ_0*m*_—initial phase of the reflected signal *m*. 

There are four multipath signals (MS1–MS4) and one direct signal (DS) illustrated in Figure 3. These quantity of signals may be larger or smaller than was drawn. All signals, which are received at point P, can have a comparable strength or some could be predominant. The direct signal is most often the strongest, but this is not always true. Such a multipath phenomenon can cause the fading of the temporary signal at point P and result in errors in the DF of a radar. The severity of multipath conditions is especially great when the so-called monopulse DF methods of microwave sources are used, and when there is no time to repeat the direction estimation of the radar. 

## 4. Monopulse Direction Finding of Microwave Sources

The monopulse direction finding of microwave emitters (such as radars) is necessary for some special applications, especially when long-term observation of emitting objects is impossible or inadvisable. The principle of monopulse DF methods relies on locating the source of emission with a single received pulse or even a select short fragment of its duration. Such systems are equipped with at least two receiving antennas. The estimated angle location φ of the microwave source is achieved based on comparing the amplitudes or phases of the signals received at each antenna. For these reasons, it can be told, respectively, about the amplitude or phase methods of direction finding of the microwave transmitters. The properties of the amplitude monopulse method (amplitude comparison monopulse) can be found in [14]. A very interesting solution of monopulse DF is described in [15]. A monopulse angle estimation based on cross-correlation operation was also presented. It is an improvement of conventional amplitude method of direction finding of microwave emitters. 

The simulations presented here were performed for the phase method of monopulse DF. 

### Phase Method of Direction Finding–General Principles

A structure of the system that enables microwave DF using the phase method, also called the interferometric method, is illustrated in Figure 4. This kind of DF system has two (or more) of the same type of omnidirectional or directional antenna A_1_ and A_2_, spaced at a distance *b* from each other. The distance *b* forms the baseline of the DF system. If directional antennas are used, the main axes of their directional patterns must be parallel, and their apertures should be placed on a common flat plane. Such a placement causes the signal levels in their output ports to be dependent on the angle φ of the microwave signal direction of arrival (DOA), as well as on their antenna gain values; however, these levels are always the same. The signal phases which are read from the output ports of antennas A_1_ and A_2_ (Figure 4), and their difference Φ are a function of the angle φ, baseline *b*, and frequency *f* of the received signals. The phase difference Φ can be measured using, for instance, the microwave phase discriminator (MPhD) [16,17,18].

The phase difference Φ of the signals at the output ports of antennas A_1_ and A_2_ can be described as follows:Φ = (2 ⋅ π ⋅ *f* ⋅ *b* / c) ⋅ sin (φ), (5)
where: *c*—velocity of light, *f*—frequency of received microwave signal, and *b*—length of baseline.

Therefore, the temporary value of the direction angle φ of the signal arrival can be estimated using Φ as evaluated with MPhD and with the frequency *f* provided by the instantaneous frequency measurement (IFM) device [19,20,21]. This is described using Equation (6) as
φ = arc sin [(Φ ⋅ *c*)/(2 ⋅ π ⋅ *f* ⋅ *b*)].(6)

More information concerning the amplitude and phase method for the monopulse DF of microwave sources is provided in references [1,2,4,22,23,24,25,26,27], for instance. The DF for various objects using the interferometric method can be performed using both electromagnetic radiation and acoustic waves [28,29]. This is because the wavelengths of acoustic waves in the audible range and of microwaves in the air are comparable. 

## 5. Results and Discussion of Simulations

All experiments should be preceded by simulations aimed at examining various features of phenomena as a function of many parameters and cases. Many effective multipath mitigation methods in the field of direction-finding have been developed using simulations [30,31,32,33,34].

In this work, the results of the simulation tests for the microwave DF device mounted on a small UAV while performing the monopulse phase method of direction finding are given as an example. The simulation environment is as follows. The basic dimensions of the simulation setup (Figure 3) are: H_R_ = 30 m, H_P_ = 7000 m, L = 40,000 m, R = 40,603 m, L_R1_ = 14,030 m, L_R2_ = 20,950 m, L_R3_ = 25,230 m. For simulation purposes, it was assumed that all multi-path signals are reflected by ground-based objects with relatively high and comparable (but not the same) values of reflectances. The structure of the DF device is presented in Figure 4. The angle of arrival φ was estimated only in the elevation plane. The algorithm of the angle φ estimation is based on Equation (6). The phase difference Φ, required in Equation (6), is evaluated as an argument (phase) of the ratio of the complex value of the resultant signal *u_A_*_2_*(t)* in the antenna A2 to the complex value of the resultant signal *u_A_*_1_*(t)* in the antenna A1. As for the resultant signals *u_A_*_2_*(t)* and *u_A_*_1_*(t)*, they are calculated taking into account the expressions (1)–(4), using the frequency *f* of the radar signal, the length of the baseline *b* (Figure 4) and the dimensions of the simulation setup (Figure 3). The assumed length of the baseline *b* is of 150 mm, and two types of signals emitted by the radar were considered. The first is the simple pulsed signal with a carrier frequency *f*_0_ of 2.8 GHz and a duration *t_i_*. The second is the complex pulsed signal with a linear frequency modulation (LFM) and duration *t_i_*. The center frequency *f*_0_ of the LFM signal is of 2.8 GHz and its deviation 2⋅Δf is 5 MHz (±2.5 MHz). In such conditions, the width of sector of the unambiguous direction angle measurement is approximately ±20 degrees. For the example, the DF antennas, which are mounted on a UAV (Figure 3), receive four signals: the direct signal is denoted as DS and the three reflected (multipath) signals are denoted as MS1, MS2, and MS3. Each reflected signal bounces off some point (object) on the ground. For the purposes of the simulations, it was assumed that multipath signals are the signals which are reflected by three ground-based objects of type passive radar decoys [35,36]. Such objects have high reflective properties, so the signals reflected from them are much stronger (dominate) than those reflected from the ground. Under such conditions ground-reflected multipath signals can be neglected. According to Figure 3, the times of arrival varied for each signal at point P.

The DS signal always reaches point P as first, and then the multipath signals MS1, MS2, and MS3 reach the point in turn. Under the considered simulation conditions, the delay time between the front slopes of the next signals was approximately 1.1 μs. The different times and angles of the signal arrivals at the point P, are the consequences of the various reflection locations for each multipath signal, as shown in Figure 3. When the pulse duration *t_i_* is shorter than the travel time of the multipath signal at point P, it is considered a short pulse case. On the other hand, when the travel time of the multipath signal at point P is shorter than the pulse duration *t_i_*, it is considered a long pulse case. The pulse duration *t_i_* assumed in the simulations is either 1 μs (short pulse case) or 4 μs (long pulse case). The temporary strength (envelopes) of the individual signals DS, MS1, MS2, and MS3 that reach antennas A_1_ and A_2_ are shown in the (a) parts of Figure 5, Figure 6, Figure 7, Figure 8, Figure 9, Figure 10, Figure 11, Figure 12, Figure 13 and Figure 14. The (b) parts of the Figure 5, Figure 6, Figure 7, Figure 8, Figure 9, Figure 10, Figure 11, Figure 12, Figure 13 and Figure 14 show the elevation angles of the signal arrivals as estimated using the two-antenna emitter location system mounted on the hypothetical UAV.

The first example are simulations for UAV1 (shown in Figure 1), which flies at a relatively high altitude. As noted in Section 2, under the conditions of UAV1, the antenna system of the DF device receives a direct signal with a considerably greater strength than the multipath signals. It was assumed in all the simulations that the angle of arrival at UAV1 for the DS is φ_DS_ = −5° and those for signals MS1, MS2, and MS3 are φ_MS1_ = −10°, φ_MS2_ = −14°, φ_MS3_ = −19°, respectively. The signal MS4 (Figure 3) was not included in the analyzes. The results of the simulations for this case are presented in Figure 5, Figure 6, Figure 7, Figure 8 and Figure 9.

The plots for the signals received by the DF antenna system installed on UAV1 for the short pulse case are presented in Figure 5a. As each pulse is received separately, the angles of their arrival are estimated precisely, as shown in Figure 5b. 

However, it is noted that only the initial elements of the plots from these figures marked as DS correctly indicate the direction of φ = −5°. The other estimated bearings shown in the figure are related with the reflected (multipath) signals MS1, MS2, and MS3, and are artefacts, i.e., their angles of arrival do not point to the radar but to the individual reflection points (objects). The same situation was observed for both the simple signal (Figure 5a) and the LFM signal emitted from the radar. In such conditions, if the mission of UAV1 is to localize the radars over an area of interest, then the plots shown in Figure 5b provide two probable hypotheses. One is that there is a detected single radar at a direction of −5°, along with received three multipath signals. The second is that four synchronously working microwave emitters (radars for instance) that emit identical signals at directions of −5°, −10°, −14°, and −19° are present. Verification of these hypotheses can be performed by comparing several observations as recorded by the DF device.

When the radar pulse duration is longer than the travel time of the multipath signals that reach the UAV (long pulse case), the DF antennas receive overlapping signals. The results for the simple signal transmitted from the radar with a duration of *t_i_* = 4 μs and for the multipath signals reflected from the ground with phases Φ_01_ = Φ_02_ = Φ_03_ = 0° are presented in Figure 6 and Figure 7. As shown in Figure 6a, the direct signal is received as an individual signal only during the first microsecond of its duration (Figure 6a and Figure 7a). During this time, the direction of the radar is estimated correctly at φ = −5°. This DF result shows the fragment of the plot denoted as DS in Figure 6b. Next, the estimated temporary values of the directions to the radar are incorrect, because the direct signal from the radar overlaps with the multipath signal MS1 at first, followed by the sum of signals MS1 and MS2, and finally by the sum of signals MS1, MS2, and MS3. 

Each fragment of the plot in Figure 6b corresponding to these situations (overlapping signals) is denoted as DS + MS1, DS + MS1 + MS2, and DS + MS1 + MS2 + MS2, respectively. After the end of direct signal, the only received signals were the multipaths, which are: the sum of MS1, MS2, MS3, the sum of MS2 and MS3, and finally the signal MS3 alone. The temporary values of the angles of arrival estimations are denoted as MS1 + MS2 + MS3, MS2 + MS3, and MS3, respectively. When assuming the signal MS2 is reflected with a phase Φ_02_ = 180°, the bearing estimated over the duration of signal MS2 (Figure 7b) resulted in values that are considerably different than that for Φ_02_ = 0° (Figure 6b). A similar situation occurs when the radar transmits an LFM signal. 

Over the period when only a single signal is received (at the starting and ending parts of the plot in Figure 8a), the temporary values of the bearings are constant. This effect is illustrated in the plots marked as DS and MS3 in Figure 8b. In the remaining parts of the graph, there were oscillations in the estimated angle of arrival φ. These oscillations are the consequence of different temporary values of the frequency from the summed signals. 

When the phase Φ_02_ of the multipath signal MS2 reflection was of 180°, there were still oscillations for the bearing values (Figure 9b), however their shapes during the MS2 duration were clearly different from those in Figure 8b. It is noted that the angles of arrival estimated during the direct signal for both the simple and LFM signal are near the direction of the radar that is φ = −5°. This is because the strength of the direct signal is greater than the power of the multipath signals. Parts a) of Figure 6, Figure 7, Figure 8 and Figure 9 contains the same plots, which is intentional for improved readability. 

Aerial vehicle UAV2 in Figure 1 flies at a relatively low altitude. It is assumed the angle of arrival was φ_DS_ = −3° for the DS and φ_MS1_ = −8°, φ_MS2_ = −12°, and φ_MS3_ = −17° for MS1, MS2, and MS3, respectively. The simulation results for UAV2 from Figure 1 are shown in Figure 10, Figure 11, Figure 12, Figure 13 and Figure 14. It is noted that in such conditions, the power for each multipath signal is clearly greater than the power of the direct signal. The conclusions from Figure 10 (short pulse case) indicate that the single radar was found correctly at a direction of −3° and the three multipath signals were separated in time and received from the angles of −8°, −12°, and −17°. Otherwise, there were four detected synchronous radars working with identical signals shifted in time and placed at directions of −3°, −8°, −12°, and −17°. 

Figure 11 and Figure 12 illustrate the long pulse cases (*t_i_* = 4 μs) when the radar transmits a simple signal. As seen, the received signals in these cases are not separated in time. The temporary overlapping of the signals causes that similarly as it was for UAV1, the results of estimation of angle of arrival does not show, nor direction of radar or directions of places of signals’ reflection. A similar phenomenon appears when the radar transmits an LFM signal. Over the period when only a single signal is received (the start of the DS and end of the MS3 part of the plot in Figure 13a), the temporary values of the bearings are constant. This is illustrated in the plots marked as DS and MS3 in Figure 13b. The remaining parts of the graph show the oscillations of the estimated angle of arrival φ. These oscillations are caused by the differences in the temporary values of the frequency for the summed signals. When the multipath signal MS2 was reflected with a phase Φ_02_ of 180°, there were still oscillations for the bearings (Figure 14b), however, their shapes for the duration of signal MS2 were clearly different than those in Figure 13b. 

The results of the simulations for UAV2 and for the long pulse cases demonstrate that the estimated values of the direction φ over the duration of signal MS2 oscillated around a value of −12°, indicating it was near the correct angle of arrival.

This effect was a result of the MS2 signal strength being greater than the other signals (DS, MS1, and MS3). Likewise, the same plots are shown as parts a) of Figure 11, Figure 12, Figure 13 and Figure 14, which was intentional for improved readability across various DF results.

## 6. Conclusions

The irregular shape of a radar antenna radiation pattern cause the efficiency of a localization of radar by means of direction finding system installed on a UAV to be dependent on the direction and height of its flightpath. The precision of the estimated angle localization of a radar is also determined by the temporary orientation and rotation velocity of the radar antenna system. If the radar antenna rotates, the moments of fading of signal received by the DF device on the UAV can occur. In these cases, the monopulse tracking system installed on the UAV can lose the radar signal, or will have an inaccurate bearings. Simultaneously, there are regularly appearing moments when the UAV is reached by the high-strength signal radiated from the main beam of the radar. At these moments, the radar will be located very precisely. Moreover, when the travel times to point P (UAV location) for the multipath signals are longer than the duration of the pulse emitted by radar, the DF system installed on the aerial vehicle receives separate signals. In these conditions, the radar direct signal is received as first at the DF system, which enables a very accurate estimation of the bearings. In turn, when the multipath signal travel times to point P are shorter than length of the radar pulse, an accurate DF of radar is still possible, but should only be based on the first portion of the received signal. 

In general, multipath phenomena increase the difficulty of DF for microwave emitters. However, this paper shows that, despite this fact, the analysis of the temporary values for the received signals and of the estimated bearings can support localizing emitter correctly by finding and rejecting false results of estimated angles of direction. As shown, even a duration of 1 μs can be enough to correctly estimate the angular location of the radar, even when multipath or jamming signals appear. However, this process needs to operate very quickly. Experiments are prepared and will be the next step of the investigations.

This paper was devoted to using microwave emission from a radar for its DF, even in a dense signal environment. This is still true when the multi-path effect does not exist but there are several radars (emitters) that work simultaneously in the area which is observed by the DF device mounted on the UAV. Only some aspects of radars DF in dense microwave environments have been shown. Many problems are still unresolved, and further research are continued. The results of the investigation may be used not only for DF of radar and other types of microwave sources, but also for localization of sonars or other objects that emit acoustic waves. Such applications could use an antenna arrays as described in [37]. The next plans concern research on the location of radars based on acoustic noise from the rotating radar antenna. 

## Figures and Tables

**Figure 1 sensors-20-05186-f001:**
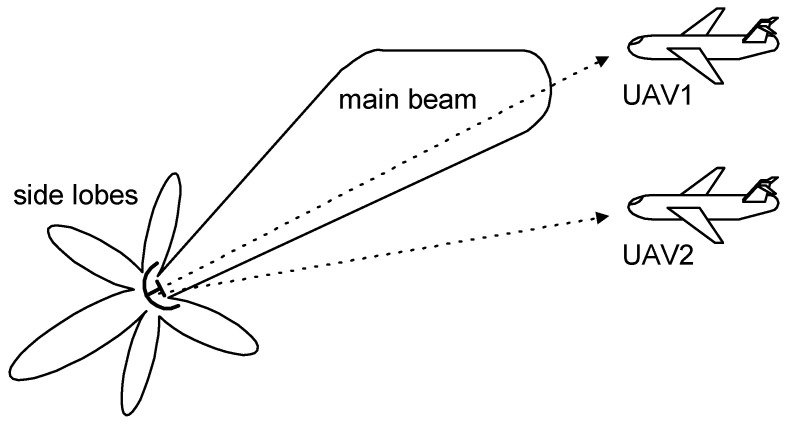
Side-view of the directional pattern for an example radar antenna system.

**Figure 2 sensors-20-05186-f002:**
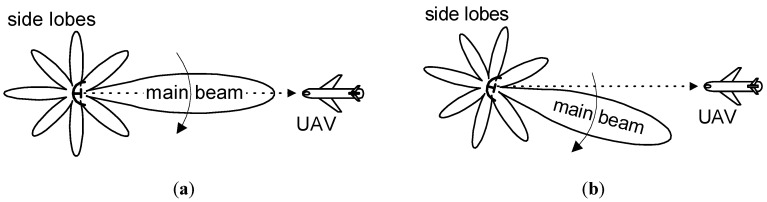
Top-view of the directional pattern for a rotating antenna showing (**a**) the main beam illuminating the unmanned aerial vehicle (UAV) at the selected moment *t*_1_ and (**b**) a minimum of the radar antenna pattern directed on the UAV at the selected moment *t*_2_.

**Figure 3 sensors-20-05186-f003:**
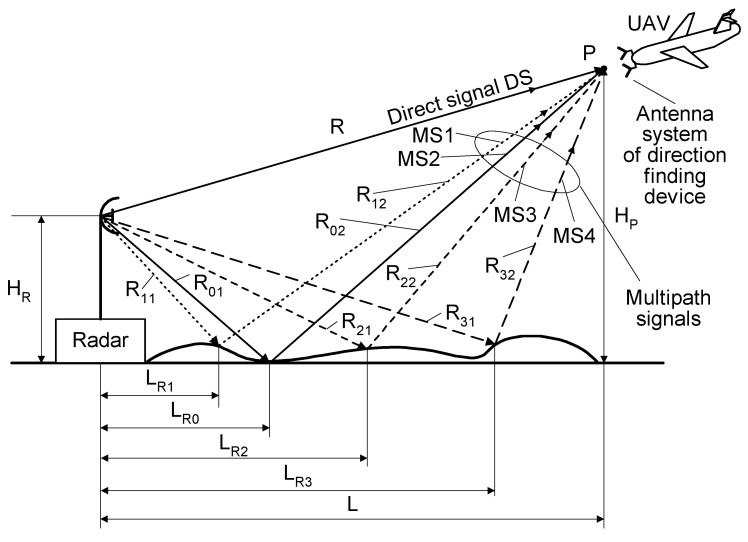
Schematic demonstrating the multipath phenomenon.

**Figure 4 sensors-20-05186-f004:**
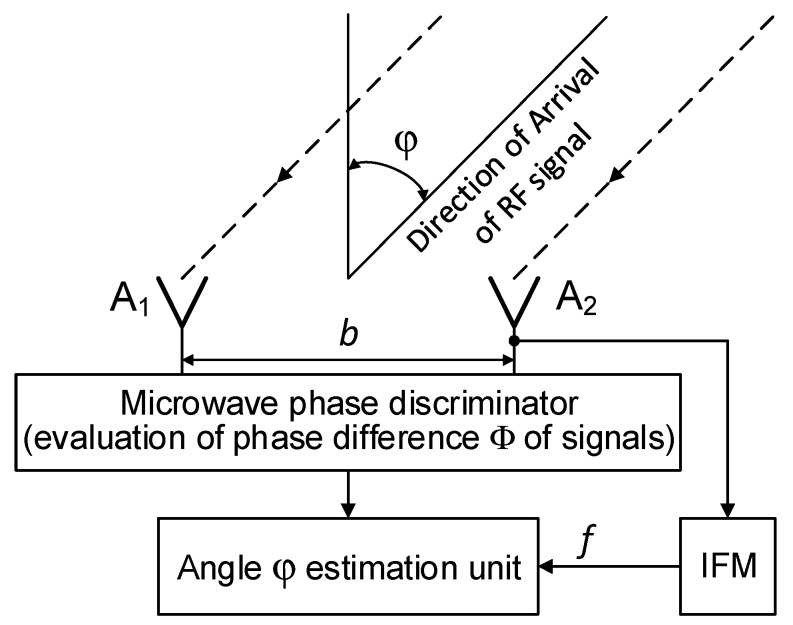
General scheme of the direction finding (DF) device using the monopulse phase method.

**Figure 5 sensors-20-05186-f005:**
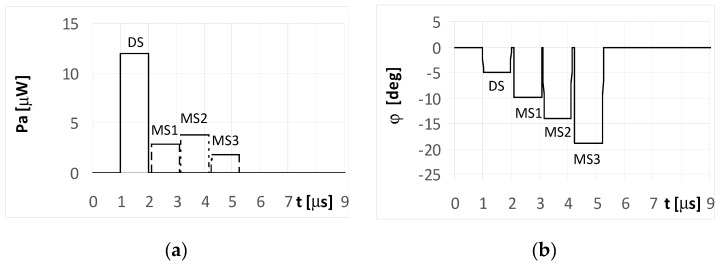
Envelopes of the direct signal (DS) and multipath signals (MS1, MS2, MS3) reflected from the ground, **separated** in time, and reaching **UAV1** (shown in Figure 1) (**a**), and temporary angle of arrival φ estimated from the received resultant signal (**b**), (for **simple and** linear frequency modulation (**LFM) short pulsed signal** as transmitted from the radar, pulse duration ***t_i_* = 1**
**μs**, and initial phases Φ_0D_ = Φ_01_ = Φ_02_ = Φ_03_ =0°; angle of arrival (AOA) was estimated using the monopulse phase method from the DF device installed on **UAV1**).

**Figure 6 sensors-20-05186-f006:**
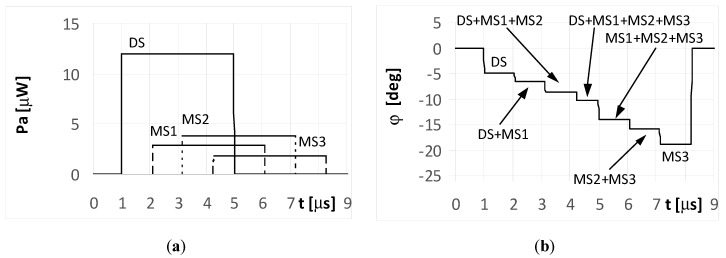
Envelopes of the direct signal (DS) and multipath signals (MS1, MS2, MS3) reflected from the ground, **overlapping** in time, and reaching **UAV1** (shown in Figure 1) (**a**), and temporary angle of arrival φ estimated from the received resultant signal (**b**), (radar transmits **simple long pulsed signal** with duration ***t_i_* = 4**
**μs**, and initial phases Φ_0D_ = Φ_01_ = Φ_02_ = Φ_03_ =0°; AOA was estimated using the monopulse phase method from the DF device installed on **UAV1**).

**Figure 7 sensors-20-05186-f007:**
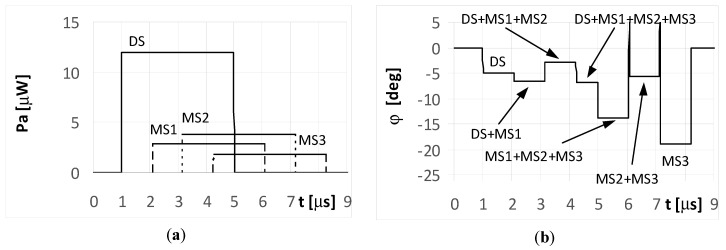
Envelopes of the direct signal (DS) and multipath signals (MS1, MS2, MS3) reflected from the ground, **overlapping** in time, and reaching **UAV1** (shown in Figure 1) (**a**), and temporary angle of arrival φ estimated from the received resultant signal (**b**), (radar transmits **simple long pulsed signal** with duration ***t_i_* = 4**
**μs**, and initial phases Φ_0D_ = Φ_01_ = Φ_03_ =0, **Φ_02_ = 180****°**; AOA was estimated using the monopulse phase method from the DF device installed on **UAV1**).

**Figure 8 sensors-20-05186-f008:**
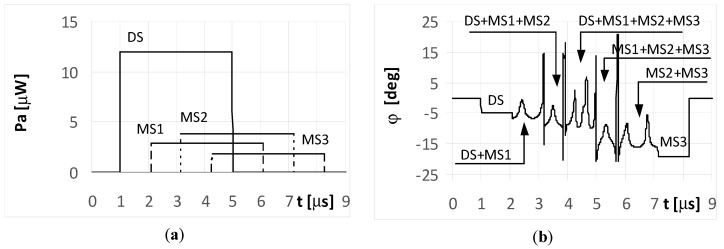
Envelopes of the direct signal (DS) and multipath signals (MS1, MS2, MS3) reflected from the ground, **overlapping** in time, and reaching **UAV1** (shown in Figure 1) (**a**), and temporary angle of arrival φ estimated from the received resultant signal (**b**), (radar transmits **LFM-type long pulsed signal** with duration ***t_i_* = 4**
**μs**, initial phases Φ_0D_ = Φ_01_ = Φ_02_ = Φ_03_ =0°, and deviation Δ*f* = 5 MHz; AOA was estimated using the monopulse phase method from the DF device installed on **UAV1**).

**Figure 9 sensors-20-05186-f009:**
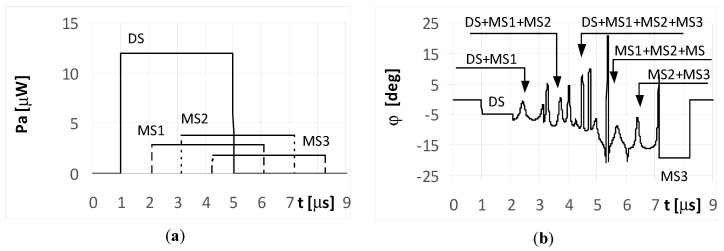
Envelopes of the direct signal (DS) and multipath signals (MS1, MS2, MS3) reflected from the ground, **overlapping** in time, and reaching **UAV1** (shown in Figure 1) (**a**), and temporary angle of arrival φ estimated from the received resultant signal (**b**), (radar transmits **LFM-type long pulsed signal** with duration ***t_i_* = 4**
**μs**, initial phases Φ_0D_ = Φ_01_ = Φ_03_ =0, **Φ_02_ = 180****°**, and deviation Δ*f* = 5 MHz; AOA was estimated using of the monopulse phase method from the DF device installed on **UAV1**).

**Figure 10 sensors-20-05186-f010:**
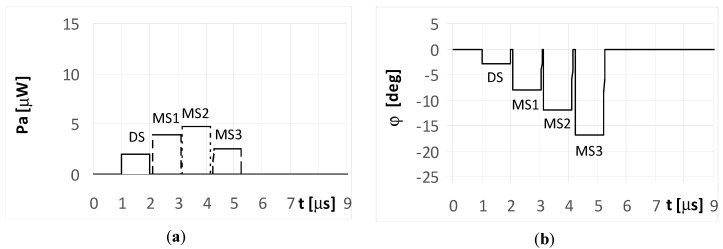
Envelopes of the direct signal (DS) and multipath signals (MS1, MS2, MS3) reflected from the ground, **separated** in time, and reaching **UAV2** (shown in Figure 1) (**a**), and temporary angle of arrival φ estimated from the received resultant signal (**b**), (for **simple and LFM short pulsed signal** transmitted by the radar, pulse duration ***t_i_* = 1**
**μs**, and initial phases Φ_0D_ = Φ_01_ = Φ_02_ = Φ_03_ =0°; AOA was estimated using the monopulse phase method from the DF device installed on **UAV2**).

**Figure 11 sensors-20-05186-f011:**
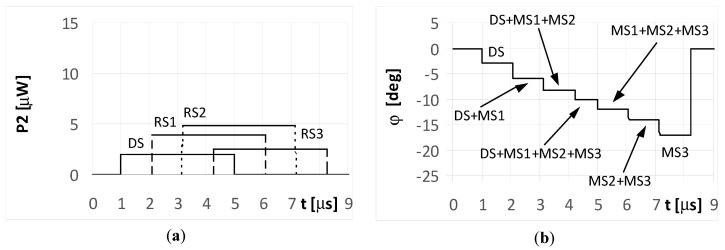
Envelopes of the direct signal (DS) and multipath signals (MS1, MS2, MS3) reflected from the ground, **overlapping** in time, and reaching **UAV2** (shown in Figure 1) (**a**), and temporary angle of arrival φ estimated from the received resultant signal (**b**), (radar transmits **simple long pulsed signal** with duration ***t_i_* = 4**
**μs**, and initial phases Φ_0D_ = Φ_01_ = Φ_02_ = Φ_03_ = 0°; AOA was estimated using the monopulse phase method from the DF device installed on **UAV2**).

**Figure 12 sensors-20-05186-f012:**
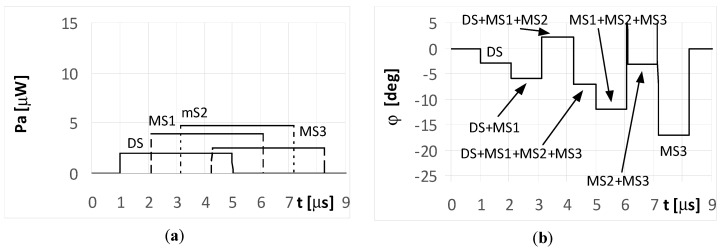
Envelopes of the direct signal (DS) and multipath signals (MS1, MS2, MS3) reflected from the ground, **overlapping** in time, and reaching **UAV2** (shown in Figure 1) (**a**), and temporary angle of arrival φ estimated from the received resultant signal (**b**), (radar transmits **simple long pulsed signal** with duration ***t_i_* = 4**
**μs**, and initial phases Φ_0D_ = Φ_01_ = Φ_03_ =0, **Φ_02_ = 180****°**; AOA was estimated using the monopulse phase method from the DF device installed on **UAV2**).

**Figure 13 sensors-20-05186-f013:**
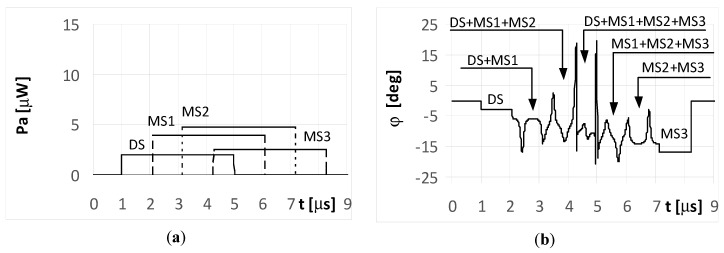
Envelopes of the direct signal (DS) and multipath signals (MS1, MS2, MS3) reflected from the ground, **overlapping** in time, and reaching **UAV2** (shown in Figure 1) (**a**), and temporary angle of arrival φ estimated from the received resultant signal (**b**), (radar transmits **LFM-type long pulsed signal** with duration ***t_i_* = 4**
**μs**, initial phases Φ_0D_ = Φ_01_ = Φ_02_ = Φ_03_ = 0°, and deviation Δ*f* = 5 MHz; AOA was estimated using the monopulse phase method from the DF device installed on **UAV2**).

**Figure 14 sensors-20-05186-f014:**
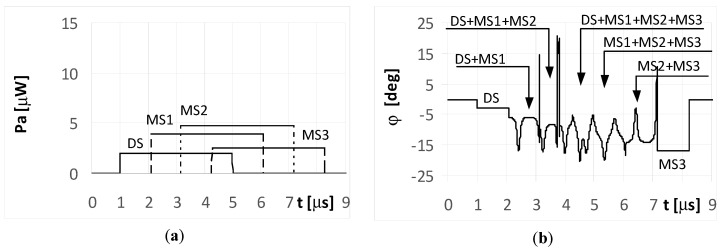
Envelopes of the direct signal (DS) and multipath signals (MS1, MS2, MS3) reflected from the ground, **overlapping** in time, and reaching **UAV2** (shown in Figure 1) (**a**), and temporary angle of arrival φ estimated from the received resultant signal (**b**), (radar transmits **LFM-type long pulsed signal** with duration ***t_i_* = 4**
**μs**, initial phases Φ_0D_ = Φ_01_ = Φ_03_ =0, **Φ_02_ = 180****°**, and deviation Δ*f* = 5 MHz; AOA was estimated using the monopulse phase method from the DF device installed on **UAV2**).

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
