# Peer review of "Some of Problems of Direction Finding of Ground-Based Radars Using Monopulse Location System Installed on Unmanned Aerial Vehicle"

_sensors, 2020, doi:10.3390/s20185186_

Round 1

Reviewer 1 Report

This manuscript is an interesting topic as a study to find the direction of the radar in a multipath fading environment. However, since the authors did not describe the background of research, related research, proposed algorithm, and its difference in more detail, it seems insufficient to be considered as a complete paper.

1. Most of the content written in the manuscript seems fundamentals, and there has a lot of research related to this topic, but the literature survey seems to be insufficient. It is necessary to find related works and emphasize the difference between what is insisted on in this manuscript.

2. In order to evaluate the performance and verify the simulation results, it is necessary to write the algorithm and simulation environment in more detail. In particular, since the authors did not describe the algorithm and the simulation environment in detail but simply showed the simulation results and explaining them, it is unlikely to realize the main algorithm the reader wants to know through the manuscript.

Author Response

Thanks once again to the Reviewer 1 for comments and suggestions.

  1. Some parts of the content of our manuscript are fundamentals indeed. It was intentionally. In this way, we wanted to clearly describe the area of our interest and draw the readers’ attention to which elements of the phenomena are of interest to us and how we try to solve our technical problems. We additional find some related works and briefly pointed out the differences between these works and our approach (lines 55-88).

  1. A description of the main elements of the direction finding algorithm which was used in simulations and a description of the simulation environment have been added (lines 207-227).

Reviewer 2 Report

This paper focuses on the direction finding of ground radars. The problem is important and timely. The authors conducted extensive experiments and provide interesting observations. Please cite and compare with the following references.

1. Yang Liu, Mugen Peng, Zhongbai Jiang, Zhiyuan Xu, Jianfeng Guan, and Su Yao, "COMP: Online Control Mechanism for Profit Maximization in Privacy-Preserving Crowdsensing", in IEEE Journal on Selected Areas in Communication, vol. 38, no. 7, pp. 1614-1628, 2020.

2. Yang Liu, Hongsheng Wang, Mugen Peng, Jianfeng Guan, Jia Xu, and Yu Wang, "DeePGA: A Privacy-Preserving Data Aggregation Game in Crowdsensing via Deep Reinforcement Learning", in IEEE Internet of Things Journal, vol. 7, no. 5, pp. 4113-4127, 2020.

3. Ruipeng Gao, Bing Zhou, Fan Ye, and Yizhou Wang, “Fast and Resilient Indoor Floor Plan Construction with a Single User,” IEEE Transactions on Mobile Computing, vol. 18, no. 5, pp. 1083-1097, 2019. 

4. Ruipeng Gao, Fangpu He, and Teng Li, “VeLoc: Finding Your Car in Indoor Parking Structures,” Sensors, 18(5):1403, 2018.

5. Ruipeng Gao, Mingmin Zhao, Tao Ye, Fan Ye, Yizhou Wang, Guojie Luo, “Smartphone-based Real Time Vehicle Tracking in Indoor Parking Structures,” IEEE Transactions on Mobile Computing, vol. 16, no. 7, pp. 2023-2036, 2017.

Author Response

Thanks once again to the Reviewer 2 for comments and suggestions.

We cited and compared three of the five references indicated by Reviewer 2 (lines 55-59). Unfortunately, our knowledge from the area of two remaining references is too little to talk about them.

Reviewer 3 Report

The authors have presented the direction finding of the ground based radar via simulation on monopulse/LFM transmission and reception and with 2 antenna system on UAV. Following are the comments:

1)the novelty and the objective of the simulation are unknown as what has been presented seems to be clear cut and intuitive. There are no need for such simulation

2)The simulation set up is too trivial wiith 1 direct path and 3 multipath. The distance between UAV and radar are unclear and so as the environment setup. Paths arrival seems to be equal strength which is not realistic.

Author Response

Thanks once again to the Reviewer 3 for comments and suggestions.

  1. In our manuscript, the main properties of the analyzed phenomena related to the direction finding are really intuitive, but in our opinion, it was necessary to perform appropriate simulations to express these properties with distinct values and better understand the specificity of these phenomena. It is difficult to find such data in available sources. For these reason, in our opinion, our work can contain elements of novelty.

  1. One direct path and only 3 multi-path tracks were considered. Such conditions were assumed in order to obtain a more clearer picture of the analyzed phenomena. Our approach can be used for more than 3 multipath or additional signals. The distance between UAV and radar and other detailed information about environment setup have been added (lines 207-227).

Round 2

Reviewer 1 Report

Since the authors reflected all the comments elaborately, I don't feel like any comments needed more.

Author Response

Thank you again to the Reviewer 1 for reviewing round 2.

We agree that the research design an methods description “can be improved”. We are working on the next paper – we hope it will be better.

There were no new detailed comments in the review report round 2.

Reviewer 3 Report

The authors has highlighted the intent with respect to the novelty and the simulation setup. For clarity and completeness, perhaps the authors can include some literature references for 

1)novelty in terms of performing appropriate simulations to express these properties with distinct values and better understand the specificity of these phenomena since it is difficult to find such data in available sources as mentioned by the authors

2) the simulation setup that related to real scenarios where one direct path and only 3 multi-path are dominant and highlighted in lines 207-227.

Author Response

Thank you again to the Reviewer 3 for reviewing round 2 and for comments and suggestions.

  1. We presented the general need for simulations in line 206-209. We have added five literature references (now lines 508-521) related to simulations in the field of direction-finding. The authors of these works searched for the properties of direction of objects estimation in the case when multipath occurs. These were multi-subject investigations. Our approach is slightly different. We worked on phase and amplitude characteristics in the time domain during a single microwave pulse. We could not find such information in these works. Of course, we will work on further aspects of monopulse angle-of-arrival estimation.

  1. We were primarily interested in cases where the multipath phenomenon is caused by some objects placed on the ground around the emitting source (eg radar). These objects can be called a radar decoy. Some explanations we added in lines 230-234. We have added two reference items (lines 522-524) for radar decoys.